# Predicting Neoadjuvant Treatment Response in Triple-Negative Breast Cancer Using Machine Learning

**DOI:** 10.3390/diagnostics14010074

**Published:** 2023-12-28

**Authors:** Shristi Bhattarai, Geetanjali Saini, Hongxiao Li, Gaurav Seth, Timothy B. Fisher, Emiel A. M. Janssen, Umay Kiraz, Jun Kong, Ritu Aneja

**Affiliations:** 1Department of Clinical and Diagnostic Sciences, School of Health Professions, University of Alabama at Birmingham, Birmingham, AL 35294, USA; bhattars@uab.edu (S.B.); gsaini2@uab.edu (G.S.); gseth@uab.edu (G.S.); 2Department of Mathematics and Statistics, Georgia State University, Atlanta, GA 30302, USA; hli35@gsu.edu; 3Department of Biology, Georgia State University, Atlanta, GA 30302, USA; tfisher10@student.gsu.edu; 4Department of Pathology, Stavanger University Hospital, 4011 Stavanger, Norway; emilius.adrianus.maria.janssen@sus.no (E.A.M.J.); umay.kiraz@sus.no (U.K.); 5Department of Chemistry, Bioscience and Environmental Engineering, Stavanger University, 4021 Stavanger, Norway

**Keywords:** triple-negative breast cancer, neoadjuvant chemotherapy, machine learning, prediction model

## Abstract

Background: Neoadjuvant chemotherapy (NAC) is the standard treatment for early-stage triple negative breast cancer (TNBC). The primary endpoint of NAC is a pathological complete response (pCR). NAC results in pCR in only 30–40% of TNBC patients. Tumor-infiltrating lymphocytes (TILs), Ki67 and phosphohistone H3 (pH3) are a few known biomarkers to predict NAC response. Currently, systematic evaluation of the combined value of these biomarkers in predicting NAC response is lacking. In this study, the predictive value of markers derived from H&E and IHC stained biopsy tissue was comprehensively evaluated using a supervised machine learning (ML)-based approach. Identifying predictive biomarkers could help guide therapeutic decisions by enabling precise stratification of TNBC patients into responders and partial or non-responders. Methods: Serial sections from core needle biopsies (*n* = 76) were stained with H&E and immunohistochemically for the Ki67 and pH3 markers, followed by whole-slide image (WSI) generation. The serial section stains in H&E stain, Ki67 and pH3 markers formed WSI triplets for each patient. The resulting WSI triplets were co-registered with H&E WSIs serving as the reference. Separate mask region-based CNN (MRCNN) models were trained with annotated H&E, Ki67 and pH3 images for detecting tumor cells, stromal and intratumoral TILs (sTILs and tTILs), Ki67^+^, and pH3^+^ cells. Top image patches with a high density of cells of interest were identified as hotspots. Best classifiers for NAC response prediction were identified by training multiple ML models and evaluating their performance by accuracy, area under curve, and confusion matrix analyses. Results: Highest prediction accuracy was achieved when hotspot regions were identified by tTIL counts and each hotspot was represented by measures of tTILs, sTILs, tumor cells, Ki67^+^, and pH3^+^ features. Regardless of the hotspot selection metric, a complementary use of multiple histological features (tTILs, sTILs) and molecular biomarkers (Ki67 and pH3) resulted in top ranked performance at the patient level. Conclusions: Overall, our results emphasize that prediction models for NAC response should be based on biomarkers in combination rather than in isolation. Our study provides compelling evidence to support the use of ML-based models to predict NAC response in patients with TNBC.

## 1. Introduction

Triple-negative breast cancer (TNBC) is an aggressive subtype of breast cancer (BC) and is often diagnosed at an advanced stage [1]. TNBC is characterized by poor prognosis and high rates of recurrence and metastasis [2]. No endocrine therapies are available for TNBC, and neoadjuvant chemotherapy (NAC) remains the mainstay of treatment for early-stage disease [3]. A small subset of patients respond to newer therapies such as poly (ADP-ribose) polymerase inhibitors and immunotherapy [4]. NAC helps to reduce tumor size before surgery [5], and its primary endpoint is a pathological complete response (pCR), defined as the absence of residual disease (RD). NAC results in pCR in only 30–40% of TNBC patients, and the remaining patients either respond moderately or are refractory to NAC (i.e., show RD) [6]. Conventional cytotoxic NAC for TNBC patients in the US consists of adriamycin, cyclophosphamide, and taxol [3]. Although the molecular basis of chemoresistance in TNBC remains elusive, inter- and intra-tumoral heterogeneity may contribute to the significant variability in NAC response observed in patients with TNBC [7].

Existing biomarkers that predict NAC response in TNBC include Ki67 and phosphohistone H3 (pH3), which capture the proliferative potential and mitotic activity of tumor cells, respectively [8,9]. Although Ki67 scoring captures the proportion of cells that have entered the cell cycle (proliferative population or P), it may not accurately represent proliferation because a Ki67-positive cell (Ki67^+^) may not divide for long periods of time [10,11]. Histone H3 is heavily phosphorylated (pH3-positive) during mitosis [12]; therefore, pH3 staining can capture true proliferation based on the identification of actively dividing cells (pH3^+^/mitotic population/M). Cells stained positive for pH3 are considered to be Ki67-positive (Ki67^+^/pH3^+^), whereas tumor cells positive for Ki67 may (Ki67^+^/pH3^+^) or may not (Ki67^+^/pH3^−^) be pH3-positive, depending on their mitotic status. Tumor assessment using hematoxylin and eosin (H&E)-stained slides of surgical resections or biopsies provides additional histological information that can predict pCR, such as the number of tumor-infiltrating lymphocytes (TILs) [13,14,15,16]. However, systematic evaluation of the ability of these biomarkers to predict NAC response has not yet been conducted.

TNBCs are complex evolving systems characterized by profound spatial and temporal heterogeneity in their biological nature and response to treatment. Individual biomarkers that depict only a single aspect of tumor physiology or biophysics are limited, and their predictive performance may vary among tumors. A multiparametric approach that combines information from functional imaging technologies with complementary sensitivity is required to predict NAC outcomes in patients with TNBC. The integration of ML-driven biomarkers into clinical practice has the potential to revolutionize cancer treatment decision making and enhance patient prognosis [17]. Thus, in this study, we developed a machine learning (ML) approach that integrates Ki67, pH3 and TILs (both intratumoral (tTILs) and stromal (sTILs)) in co-registered serial whole-slide images (WSIs) to predict NAC response in patients with TNBC. Such a computational system that jointly uses H&E and IHC biomarkers from serial tissue slides for predicting NAC response serves a promising avenue for treatment optimization and customization, a necessary step towards the realization of personalized medicine.

## 2. Materials and Methods

### 2.1. Study Cohort

A total of 76 formalin-fixed paraffin-embedded (FFPE) TNBC biopsy samples were retrieved from Emory Decatur Hospital. Biopsy samples were collected before systemic treatment. Of these patients, 44 showed pCR, and the remaining 32 showed RD after NAC treatment. All study aspects, including study protocols, sample procurement, and study design, were approved by the Institutional Review Board (IRB). Clinicopathological data, patient survival information, and NAC response data were available (Appendix A). For each patient, three tissue slides containing serial sections (5 μm) were stained with H&E, Ki67, and pH3.

### 2.2. Serial Image Co-Registration

High-resolution WSIs of three serial tissue slides were produced for each patient; one was stained with H&E and two were immunohistochemically stained for Ki67 and pH3. WSI triplets were co-registered at the highest image resolution using our previously developed dynamic co-registration method [18]. For each image triplet, the H&E WSI served as the reference image and the other two IHC images were mapped to the reference image. After image registration, our dataset consisted of 1044 WSI image regions of 8000 × 8000 pixels from 76 TNBC patients. As the registered images were large in size and Graphical Processing Units (GPUs) for model training and testing have limited memory, each image region was further partitioned into non-overlapping image patches of 1000 × 1000 pixels by size. The resulting image patches were appropriate for ML analyses.

### 2.3. H&E Staining

FFPE samples were subjected to a series of xylene (X5-4, Fisher Scientific, MA, USA) washes followed by alcohol (BP2818-4, Fisher Scientific, MA, USA) washes. The tissues were then thoroughly rinsed with water and stained with Hematoxylin (CATHE-MM, Biocare Medical, CA, USA). Subsequently, tissue sections were stained with Eosin (HT11-132, Sigma Aldrich, MA, USA), which stains nonnuclear elements in different shades of pink. After rinsing in a series of alcohol solutions and xylene, a thin layer of polystyrene mountant (022-208, Fisher Scientific, MA, USA) was applied, followed by tissue mounting on a glass coverslip (Globe Scientific Inc., NJ, USA).

### 2.4. Immunohistochemistry

Immunohistochemistry (IHC) was performed as previously described [18]. FFPE tissue sections were deparaffinized by a 20 min incubation in an oven, followed by a series of xylene washes. The tissues were rehydrated in a series of ethanol solutions (100%, 90%, 75%, and 50%). Antigen retrieval was performed by heating tissues in a citrate buffer (pH 6.0) using a pressure cooker (15 psi) for 30 min. The tissues were cooled to room temperature and then incubated in hydrogen peroxide for 10 min, followed by blocking in UltraVision Protein Block (Life Sciences Inc., St. Petersburg, FL, USA) for 10 min. Tissue samples were incubated for 60 min at room temperature with primary antibodies (Monoclonal Mouse Anti-Human Ki67, clone MIB-1, Dako North America Inc., DeLand, FL, USA at 1:100 dilution; Phosphohistone H3 (pH3), Biocare Medical, Pacheco, CA, USA, 1:500 dilution). After a series of washes, tissues were incubated with a MACH2 HRP-conjugated secondary antibody (Biocare Medical, Pacheco, CA, USA). Enzymatic antibody detection was performed using Betazoid DAB Chromogen Kit (Biocare Medical, Pacheco, CA, USA). Finally, the tissue sections were counterstained with Mayer’s hematoxylin, dehydrated in a series of ethanol concentrations, and mounted with mounting media.

### 2.5. Imaging and Tumor Slide Annotation

Slides were scanned using a slide scanner (Hamamatsu NanoZoomer 2.0-HT C9600-13 (Hamamatsu, Japan)) at 40× magnification (0.23 μm/pixel). Using Aperio ImageScope 12.4.3, board-certified pathologists (EJ and UK) reviewed the images for quality, overlapping tissue areas, out-of-focus areas, and staining artifacts. Tumor cells, sTILs and tTILs were manually annotated in H&E WSIs. According to international guidelines [19], tTILs were defined as lymphocytes in a tumor cell nest in direct contact with adjacent tumor cells, and sTILs were defined as lymphocytes in the tumor stroma and not in direct contact with tumor cells. Artifacts and necrotic areas were manually excluded from evaluation.

### 2.6. ML Pipeline

To emulate a pathologist’s review process, we developed a traditional ML prediction pipeline (Figure 1) consisting of multiple processing steps, including serial histopathological image generation (Appendix A), registration, cell detection, biomarker identification, hotspot region detection, hotspot feature extraction, classifier optimization, and NAC response prediction.

### 2.7. Cell Detection and Phenotyping

Due to its promising performance and strong support for different analysis tasks (e.g., detection and segmentation), separate Mask R-CNN (MRCNN) models [20] were used to detect tumor cells, TILs, and cells stained with IHC biomarkers in spatially aligned WSIs. The MRCNN model for tumor cell detection was trained using 797 H&E images of 1000 × 1000 pixels with all tumor cells annotated by expert pathologists. The MRCNN model for TIL detection was trained using 500 H&E images of 1000 × 1000 pixels with TIL annotations by pathologists. In contrast, the MRCNN models for Ki67^+^ and pH3^+^ tumor cell detection were trained using 30 and 20 well-annotated IHC images (1000 × 1000 pixels), respectively. The details of individual training datasets are given in Table 1. Cells of interest for NAC response prediction included tumor cells, sTILs, tTILs, Ki67^+^, and pH3^+^. sTILs and tTILs were marked by pathologists, whereas Ki67^+^ and pH3^+^ were identified by mapping serial IHC images to the reference H&E image (Figure 1A). Tumor cells in H&E images were identified as Ki67^+^ or pH3^+^ if Ki67 or pH3 staining was detected within a radius of 40 pixels. This radius was empirically determined by taking the average tumor cell size of cells (>1 million) from more than 1000 images of 1000 × 1000 pixels by size.

### 2.8. Identification of Hotspot Regions

We identified hotspot regions enriched with cells of interest in WSIs. Cell types of interest for hotspot recognition included sTILs, tTILs, sTILs + tTILs, Ki67^+^ (henceforth referred to as P), pH3^+^ (henceforth referred to as M), and tumor cells (TCs). For each selection metric, all image patches from each patient were sorted in descending order, and the top image patches with a high cell density were selected as hotspots. A typical hotspot selected by pathologists includes 500–2000 tumor cells [21,22]. Given that each image patch in our dataset included an average of 100 tumor cells, we included a sufficiently large number of tumor cells by considering the top 21 image patches as hotspots. The odd hotspot number made it easier to support majority voting while deriving the predicted patient class label from hotspot labels. As each image patch has the same image size, the cell count is equivalent to the cell density. We considered the number of cells (i.e., 21 × 100 = 2100) large enough to represent different cell populations in each tissue slide, which is comparable to the manual pathology review process. To make the hotspot selection process sufficiently comprehensive, a diverse set of selection criteria was used, including selection by counts of tTILs, sTILs, sTILs + tTILs, M, P, and TCs.

### 2.9. Classifier Optimization and NAC Response Prediction

The NAC response predictor was trained using cell profile features extracted from the hotspot regions. Features derived from each hotspot included counts of sTILs, tTILs, M, P, and TCs. All hotspots from the same patient shared the same patient-class label (i.e., either pCR or RD). To identify the best classifiers for NAC response prediction, we used multiple ML methods, including linear discriminant analysis (LDA), support vector machine (SVM), and multilayer perceptron (MLP). We optimized the hyperparameters of these ML algorithms using the auto-ML tool Auto-Sklearn [23]. The details of model parameters are provided in Appendix A.

The resulting hotspot dataset included 1596 image patches from 76 patients (44 with pCR, and 32 with RD) and was randomly split into 75% for training (57 patients: 33 with pCR, and 24 with RD) and 25% for testing (19 patients: 11 with pCR, and 8 with RD). Prediction performance was evaluated using a three-fold cross-validation method. The trained model with the highest average validation accuracy was retained and applied to the testing data. The image patch-level results were aggregated for patient-level prediction results using the maximum voting rule. For analysis pipeline implementation, pCR was defined as the positive class and RD as the negative class.

Prediction accuracy was defined as
(1)$$Accuracy=\frac{TP+TN}{TP+FN+TN+FP},$$
where *TP*, *FP*, *TN*, and *FN* represent true positive, false positive, true negative, and false negative, respectively.

The true positive rate (*TPR*) and the false positive rate (*FPR*) were defined as
(2)$$TPR=\frac{TP}{TP+FN},$$
(3)$$FPR=\frac{FP}{FP+TN}.$$

Furthermore, we assessed prediction accuracy using the area under the receiver operating characteristic curve (ROC AUC). The ROC analysis involved plotting the true positive rate against the false positive rate to evaluate the diagnostic capacity of the classifiers. The ROC AUC serves as a measure of the classifier’s capability to differentiate between positive and negative cases [24].

## 3. Results

### 3.1. Prediction Accuracy of ML Classifiers for Different Hotspots

The prediction accuracies and F1 scores of the optimal ML classifiers for hotspots by different selection criteria (i.e., tTILs, sTILs, sTILs + tTILs, M, P, TCs) are presented in Figure 2 where each hotspot is represented by tTILs, sTILs, P, M, and TCs. At the patient level, the highest prediction accuracy was achieved with tTIL classifiers. In contrast, patient-level prediction accuracy for sTIL hotspots was relatively low. When sTILs and tTILs were jointly considered (sTILs + tTILs) as hotspots, the resulting prediction performance lay between the prediction accuracies of the individual hotspots. In general, the prediction accuracy at the image patch level was lower than that at the patient level. Because of the prediction robustness introduced by the maximum voting mechanism, patient-level prediction performance was not significantly affected by misclassification rates at the image patch level. The patient-level prediction accuracy for M or P hotspots was lower than that for TIL hotspots. Although patient-level prediction accuracy was higher for P hotspots than for M hotspots, the opposite trend was observed for image patch-level prediction accuracy. F1 score results suggested similar observations (Figure 2).

Receiver operating characteristic (ROC) curve analysis was conducted to calculate the area under the curve (AUC) values for the different prediction models (Figure 3). Prediction performance for tTIL hotspots was the highest among all hotspots, demonstrating the considerable impact of hotspot selection standard on prediction results. When tTILs and sTILs were jointly considered for hotspot selection, the AUC value was higher than the AUC value obtained with the model for sTIL hotspot prediction (Figure 3). Moreover, the AUC value for M hotspots was higher than that for P hotspots, but lower than that of the TC hotspots.

### 3.2. Biomarker Hotspots Accurately Predict pCR in Patients with TNBC

The patient- and image patch-level testing performances of the optimal prediction models for different hotspots were compared using confusion matrix analysis (Figure 4 and Figure 5). Regardless of the hotspot standard, all prediction models correctly identified all patients with pCR (except the sTIL hotspots model that misclassified one case) with no false-negative results at the patient level. However, different hotspot detection models showed variable RD prediction performance. The misclassification rate for RD exhibited considerable variability, ranging from 50% when considering tTIL hotspots to reaching a maximum of 100% in cases for M or TC hotspots. These findings remained consistent when we extended our analysis to the image patch level, mirroring the patterns observed at the hotspot level (Figure 5). Importantly, the image patch-level prediction accuracy for pCR reached 92% with tTIL hotspots.

### 3.3. Prediction Accuracy Associated with Different Hotspot Features

We investigated the best prediction accuracies for hotspot features at the image patch and patient levels using different hotspot selection metrics (Table 2). Overall, hotspots identified by tTIL counts (i.e., tTILs hotspots) showed the best prediction accuracy at the patient level. The combination of tTIL count and additional cell counts (sTILs, P, and M) enhanced the prediction accuracy at both the image patch and patient levels. Similarly, for the other hotspot selection metrics, the use of multiple cellular features (i.e., counts of tTILs, sTILs, and TCs) and molecular biomarkers (i.e., P and M) resulted in improved prediction performance, suggesting the presence of complementary prediction values from these classes of hotspot features (H&E- and IHC-derived data).

## 4. Discussion

pCR is associated with favorable prognosis in neoadjuvant trials in TNBC and is considered a surrogate marker of disease-free survival and overall survival (OS) [25]. A comprehensive assessment of the predictive value of potential biomarkers for pCR (Ki67, pH3, TILs) in patients with TNBC can guide therapeutic decision making by enabling precise stratification of patients into responders and partial or non-responders who may be spared the unnecessary adverse effects of NAC and directly receive surgery. This could also help clinicians identify immunomodulatory strategies for partial responders, which, in combination with chemotherapy, may enhance the response to NAC and improve treatment outcomes. In this study, we comprehensively evaluated the predictive value of markers derived from H&E and IHC stained biopsy tissue using a supervised ML-based approach. In spite of the promising success in various biomedical image classification studies, deep learning does not achieve a satisfying interpretability overall. Additionally, automatically extracted features may not be biologically meaningful with an imminent translational impact. To address these limitations, deep learning models only for cell and biomarker counting and pre-defined biologically meaningful features for NAC response prediction were used. ML algorithms were applied to the spatially aligned H&E and IHC images to detect Ki67, pH3, and TIL hotspots since ML methods can outperform human reviewers in terms of accuracy, speed, and reproducibility [26]. They can also reveal predictive morphological features and spatial patterns beyond human perceptual abilities [27,28].

Our findings suggest that the hotspot selection metric significantly affects the prediction performance of the model. Notably, TILs (derived from H&E-stained tissue slides) exhibited strong predictive accuracy, outperforming P (Ki67^+^, derived from IHC stained tissue sides), which is the current clinical standard. Our data suggest that the tTILs count is the optimal standard for hotspot selection which is in line with published studies on the pCR predictive value of TILs. H&E staining is the gold standard for histopathological assessment of the tumor microenvironment (TME) [29]. H&E-stained WSIs provide vital information about tissue architecture, including the type of cells (e.g., epithelial, stromal, and TILs) and their spatial arrangement in the dynamic TME. Several studies have shown that histopathological TME components, particularly TILs, can be used to predict pCR because increased counts of sTILs and tTILs have been correlated with pCR in TNBC patients [30,31]. TILs mirror the local immune response, and high TIL counts are associated with favorable OS in patients with TNBC [32,33,34]. Although H&E WSIs can be used to enumerate TILs, they do not provide information on the various types of TILs in the TME. For example, patients with tumors enriched in TILs may have poor prognoses, possibly due to abundance of protumoral over antitumoral immune cells, presence of exhausted immune cells, or unfavorable positioning of TILs. Techniques such as imaging mass cytometry allow for an in-depth characterization of the immune TME by distinguishing between the various TIL subtypes (e.g., T-cells, B-cells, macrophages, etc.) and furnishing their spatial information. A better understanding of the immune landscape can uncover the connections between immune components and therapeutic response, yielding novel predictive markers for NAC response in TNBC.

After hotspot identification, we used different combinations of cellular features (counts of tTILs, sTILs, and TCs) and molecular biomarkers (Ki67 and pH3) to predict NAC response. The counts of sTILs, tTILs, P, and M, considered in tTIL hotspots, exhibited the highest predictive value. The combination of histological features from H&E images and molecular biomarkers from IHC images performed better than individual features, supporting the complementary predictive ability of these two classes of features. Overall, our results emphasize that prediction models for NAC response should be based on biomarkers in combination rather than in isolation.

Tumors with a higher level of cell proliferation respond better to chemotherapy than tumors with a lower level of proliferation [35,36]. The Ki67 index is a routinely used clinical marker of cancer cell proliferation and is strongly correlated with recurrence and metastasis [37,38,39]. Nuclear Ki67 is expressed in all active phases of the cell cycle (G1, S, and G2) [40,41]. Patients with high Ki67 levels respond well to NAC [42], and thus, Ki67 levels can independently predict pCR. However, the cutoff value for classifying Ki67 expression as high or low is not standardized and ranges from 12% to 25% [43]. Ki67 may fail to accurately represent actively dividing cells. Although Ki67 scoring captures the proportion of cells that have entered the cell cycle, it may misrepresent proliferation in the truest sense, as Ki67^+^ cells (proliferating or P population) may not divide for long periods of time, and cells in the G1 phase have uncertain fates [44,45,46,47]. Current clinical Ki67 evaluation is highly subjective, causing differing opinions among pathologists while selecting fields for assessment of heterogeneous tumors, such as TNBC. Chemotherapy targets actively dividing cells during mitosis; hence, pH3, a marker of mitotic activity, has emerged as a predictor of NAC response. Histone H3 is a core histone protein, and its phosphorylation occurs exclusively during mitosis (S-phase) [12,48]; therefore, pH3 staining can capture true proliferation based on the identification of mitotic cells (M population). Although pH3 staining has not yet been clinically adopted, its ability to predict pCR has been demonstrated in several clinical studies, and it is superior to Ki67 staining in terms of reproducibility and ability to represent proliferation [12]. In current practice, the P (Ki67^+^) and M (pH3^+^) populations of tumor cells are evaluated independently from a limited number of high-power tumor fields [49] and the predictive power of either variable alone may result in inaccurate patient risk stratification or clinical decision making. Therefore, Ki67 should be considered in conjunction with pH3, in the same microscopic field or region of the tumor, to obtain a true picture of tumor cell proliferation. Traditional scoring methods for Ki67 and pH3 are time-consuming and prone to inter- and intra-observer variability, limiting the clinical value of these biomarkers. By contrast, ML methods can process entire slides for multiple markers concurrently, yielding enriched information in a fraction of the time. ML is less susceptible to inter- and intra-observer subjectivity than manual scoring and can identify novel predictive features and spatial patterns. Thus, ML-based prediction models present immense clinical potential and are poised to become mainstream tools for breast cancer diagnosis. By leveraging co-registered H&E and IHC WSIs of serial tissue sections, our ML-based pipeline integrates complementary information from histological components (TCs, tTILs, and sTILs in H&E slides) and biomarkers (Ki67 and pH3 in serial IHC slides) to predict NAC response. All components in this computational pipeline are fully automated, making it more efficient than traditional manual reviewing. Our findings suggest that the prediction accuracy from feature P alone, considered within the P hotspot identification metric, is better than that from feature M alone within the M hotspot identification metric, both at the image patch and patient level. Furthermore, a joint consideration of the features—sTILs, tTILs, M, P, and TCs—resulted in improved prediction accuracy at the image patch and patient levels for M and P hotspot identification metrics. Thus, considering both sets of populations, the actively dividing one and the subset that is not, may yield improved predictive value.

This study has some limitations. The sample size in this study is relatively small, partially due to the fact that three serial tissue slides have to be produced and stained for each patient. The biopsy blocks were inadequate for producing three consecutive sections. In our pipeline, we considered select TME components (tumor cells, tTILs, sTILs,) and biomarkers (Ki67 and pH3). However, the pipeline can be extended to include a larger number of predictive biomarkers and clinicopathological data (e.g., age, tumor grade, stage, and lymph node status) to enhance the predictive ability of the model. Additionally, hotspots can be detected using different metrics and fixed cutoff values rather than using dynamic cutoffs determined by the data. These modifications may better mirror current clinical practice protocols and enhance the interpretability of NAC response prediction. Furthermore, image patch-level results were aggregated with patient-level results using the maximum voting rule. By doing so, we implicitly assumed the equal impact of all image patches on patient-level prediction results. We plan to optimize this aggregation rule at a better resolution in future research. Due to limitations of immunohistochemistry, a subset of the M population may not be positively stained for Ki67 (appearing erroneously as pH3^+^/Ki67^−^), and similarly, a subset of the P population that is in the mitotic cycle (should be Ki67^+^/pH3^+^) may not be positively stained for pH3 (appearing as Ki67^+^/pH3^−^). Imprecise co-registration can also result in a subpopulation of M cells lacking a Ki67signal (pH3^+^/Ki67^−^).

## 5. Conclusions

Our study provides compelling evidence to support the use of ML-based models to predict NAC response in patients with TNBC. With its effectiveness and interpretability, our prediction model shows a promising potential for clinical adoption in future.

## Figures and Tables

**Figure 1 diagnostics-14-00074-f001:**
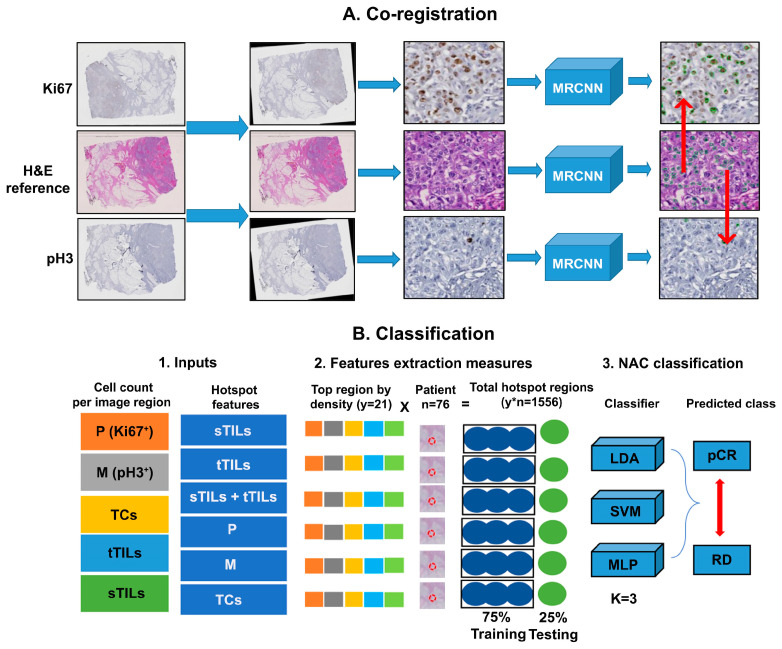
Schema of our ML pipeline. (**A**) Co-registration—three WSIs (H&E, IHC-Ki67, and IHC-pH3) for each patient were acquired and spatially registered at the highest image resolution before being partitioned into image patches of 1000 × 1000 pixels. Multiple MRCNN models were trained with independent labelled datasets and applied to image patches to detect TCs (tumor cells), TILs (tumor infiltrating lymphocytes) in H&E slides, P (tumor cells stained positive for Ki67) and M (tumor cells stained positive for pH3) in IHC images. (**B**) Classification—this part of the pipeline included inputs of image patches and cell counts. The image patches with the highest spatial density of cells of interest were identified as diagnostic hotspots. The numbers of TCs, P, M, sTILs, and tTILs from the hotspots were used as hotspot features to train (75%) and test (25%) the classification models for NAC response prediction.

**Figure 2 diagnostics-14-00074-f002:**
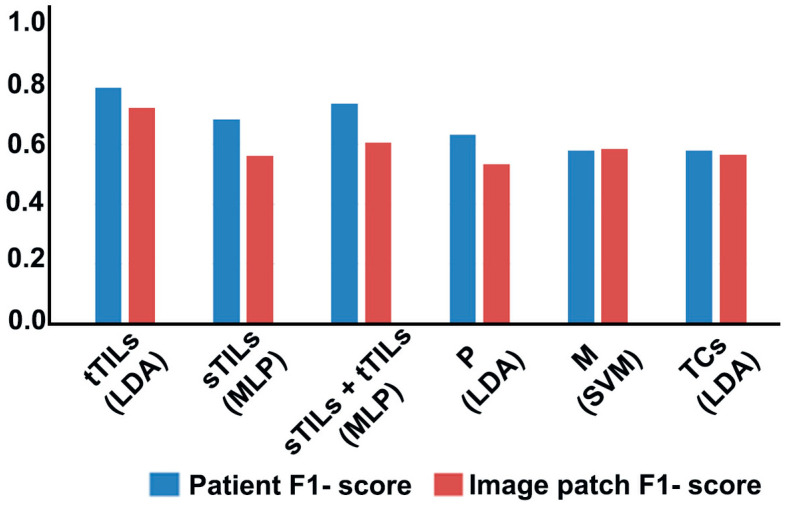
Prediction accuracies and F1 scores of the optimal ML classifiers using different hotspot selection standards with hotspot feature representation by tTILs, sTILs, P, M, and TCs. In contrast, MLP showed the best performance when the count of sTILs or sTILs + tTILs was used as hotspot selection standard. SVM performed best when M was used as the hotspot selection standard. LDA = linear discriminant analysis; SVM = support vector machine; MLP = multilayer perceptron.

**Figure 3 diagnostics-14-00074-f003:**
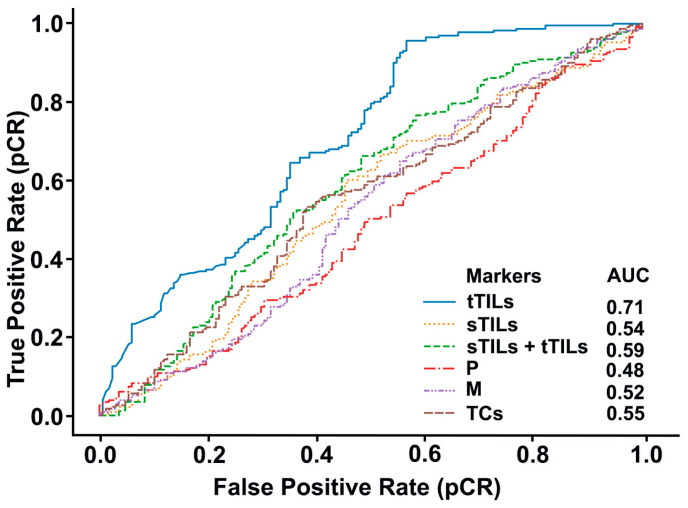
ROC curves of the best classifiers according to different hotspot selection standards. Best classifier performance was observed in tTILs (AUC = 0.71), followed by sTILs + tTILs (AUC = 0.59).

**Figure 4 diagnostics-14-00074-f004:**
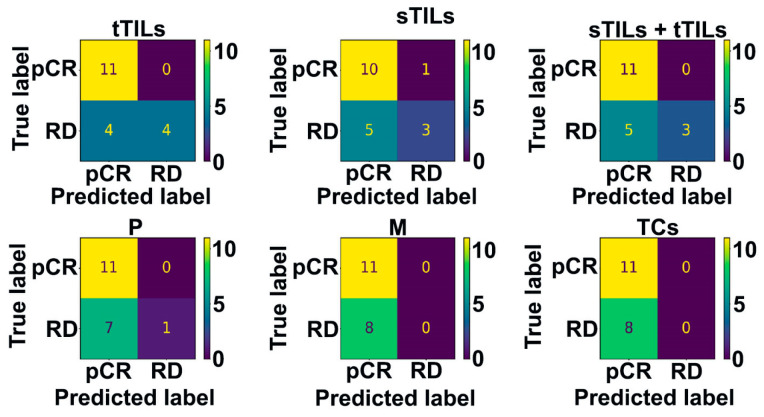
Confusion matrices showing patient-level testing performance. When hotspots were identified by the count of tTILs, sTILs, sTILs + tTILs, P, M, and TCs, the associated best classifiers showed different performances; pCR prediction performance was accurate, with variable RD prediction performance. Patient-level prediction label was determined using patch-level labels by maximum voting.

**Figure 5 diagnostics-14-00074-f005:**
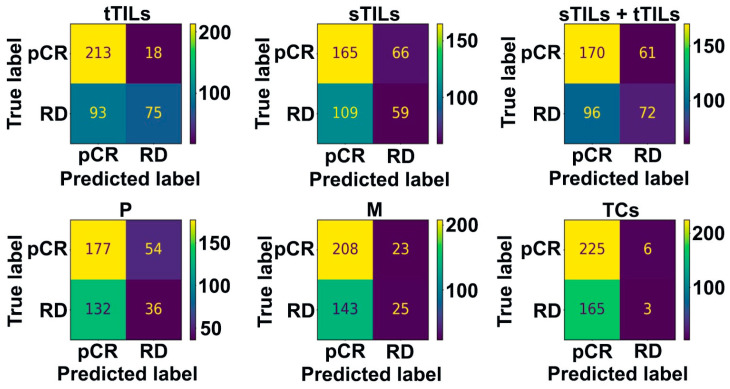
Confusion matrices showing image patch-level testing performance. When the hotspots were identified by the count of tTILs, sTILs, sTILs + tTILs, P, M, and TCs, the associated best classifiers showed different performances as suggested by the patch-level confusion matrices with values normalized by rows, while pCR prediction accuracy was high (92% with tTILs hotspots), the RD prediction accuracy was variable.

**Table 1 diagnostics-14-00074-t001:** The details of individual training datasets for separate MRCNN models.

Model Function	Image Stain	Image Count	Image Size	Annotation Type	Annotated Cell Type	Annotated Cell Number
Tumor cell detection	H&E	797	1000 × 1000 pixels	Point	Tumor cell	55,506
TIL detection	H&E	500	1000 × 1000 pixels	Point	TIL	26,415
Ki67^+^ tumor cell detection	Ki67	30	1000 × 1000 pixels	Contour	Ki67^+^ tumor cell	856
pH3^+^ tumor cell detection	pH3	20	1000 × 1000 pixels	Contour	pH3^+^ tumor cell	37

**Table 2 diagnostics-14-00074-t002:** The accuracies of hotspots with various feature combinations using different ML algorithms. Note that the metrics of “Precision”, “Recall”, and “F1-Score” are calculated with “pCR” being the positive class.

Hotspot Identification Metric	Features Considered in Hotspots	Image Patch Accuracy	Patient Accuracy	ML Algorithm	Image Precision	Image Recall	Image F1-Score	Patient Precision	Patient Recall	Patient F1-Score
TCs	TCs	0.579	0.579	LDA	0.579	1.000	0.733	0.579	1.000	0.733
	tTILs; sTILs; P; M; TCs	0.566	0.579	LDA	0.577	0.974	0.725	0.579	1.000	0.733
	tTILs; sTILs; P; M	0.576	0.579	SVM	0.579	1.000	0.733	0.579	1.000	0.733
tTILs	tTILs	0.702	0.684	LDA	0.663	0.987	0.793	0.647	1.000	0.786
	tTILs; sTILs; P; M; TCs	0.722	0.789	LDA	0.696	0.922	0.793	0.733	1.000	0.846
	tTILs; sTILs; P; M	0.739	0.789	LDA	0.702	0.957	0.810	0.733	1.000	0.846
sTILs	sTILs	0.634	0.632	LDA	0.617	0.970	0.754	0.611	1.000	0.759
	tTILs; sTILs; P; M; TCs	0.561	0.684	MLP	0.602	0.714	0.653	0.667	0.909	0.769
	tTILs; sTILs; P; M	0.632	0.632	LDA	0.618	0.952	0.750	0.611	1.000	0.759
sTILs + tTILs	tTILs; sTILs	0.657	0.632	LDA	0.634	0.965	0.765	0.611	1.000	0.759
	tTILs; sTILs; P; M; TCs	0.607	0.737	MLP	0.639	0.736	0.684	0.688	1.000	0.815
	tTILs; sTILs; P; M	0.632	0.632	LDA	0.618	0.952	0.750	0.611	1.000	0.759
P	P	0.534	0.526	MLP	0.570	0.797	0.664	0.563	0.818	0.667
	tTILs; sTILs; P; M; TCs	0.534	0.632	LDA	0.573	0.766	0.656	0.611	1.000	0.759
	tTILs; sTILs; P; M	0.574	0.526	LDA	0.589	0.874	0.704	0.556	0.909	0.690
M	M	0.499	0.421	LDA	0.549	0.749	0.634	0.500	0.727	0.593
	tTILs; sTILs; P; M; TCs	0.584	0.579	SVM	0.593	0.900	0.715	0.579	1.000	0.733
	tTILs; sTILs; P; M	0.599	0.579	LDA	0.601	0.918	0.726	0.579	1.000	0.733

TCs = Tumor cells; tTILs = intratumoral TILs; sTILs = stromal TILs; P = Ki67^+^; M = pH3^+^; LDA = linear discriminant analysis; SVM = support vector machine; MLP = multilayer perceptron; ML = machine learning.

## Data Availability

The data underlying this article will be shared on request to the corresponding author.

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
