# Peer review of "Predicting Neoadjuvant Treatment Response in Triple-Negative Breast Cancer Using Machine Learning"

_diagnostics, 2023, doi:10.3390/diagnostics14010074_

Round 1

Reviewer 1 Report

Comments and Suggestions for Authors

In this paper, the authors proposed an interesting machine learning pipeline to predict the triple negative breast cancer patient (TNBC) response to neoadjuvant chemotherapy (NAC) using biomarkers generated from tissue slides. The paper is well organized. I have a few suggestions and questions as follows.

Q1: In section "Cell detection and phenotyping", the authors clearly mentioned that cell count model by MRCNN is pre-specified, i.e., it is generated from other independent data sets. Can the authors also clearly stated it in Figure 1 notes? Otherwise, after a quick look of Figure 1, readers may misunderstand that the input in classification has contained some information about pCR / RD to NAC, i.e. the response variable is MRCNN is pCR / RD to NAC.

Q2: In classification, why the authors select top 21 hotspot regions? To my understanding, it is also a hyper-tuning parameter in this machine learning pipeline. The authors should try different number of top hotspot regions, i.e., 10, 20 and 30, to see if the predictive performance is stable and when the predictive performance is best.

Q3: Can the authors explain why try LDA, SVM and MLP in classification? How about also trying some widely-used machine learning model that may work well for small data set such as logistic regression, regression tree and gradient boosting?

Q4: Generally speaking, the sample size in this paper is relatively small. Only 76 patients are included. Do the authors have any comments about the findings reproducibility?

Author Response

Dear Reviewer,

Thank you for giving us the opportunity to revise and resubmit our manuscript entitled “Predicting neoadjuvant treatment response in triple-negative breast cancer using machine learning” Manuscript ID: diagnostics-2726165 in Diagnostics. The comments were overall very positive and encouraging. Please see the attachment for a point-by-point response to your suggestions and the revised manuscript.

Thank you

Reviewer 2 Report

Comments and Suggestions for Authors

see the report

Author Response

(The authors gave the same response as above.)

Round 2

Reviewer 1 Report

Comments and Suggestions for Authors

Thanks for the authors' reply. For the question 2 and 3, it is better to use numbers to justify. Please try some other tuning parameters (number of top hotspot regions) and some other popular machine learning methods, and use the numbers to justify current  findings / model structures is most robust. 

Author Response

Dear Reviewer

       Thank you for giving us the opportunity to revise and resubmit our manuscript entitled “Predicting neoadjuvant treatment response in triple-negative breast cancer using machine learning” Manuscript ID: diagnostics-2726165 in Diagnostics. Please find the attached file with point-by-point response to your valuable suggestions.

Thank you!
